# Experimental Study on the Shrinkage Behavior and Mechanical Properties of AAM Mortar Mixed with CSA Expansive Additive

**DOI:** 10.3390/ma12203312

**Published:** 2019-10-11

**Authors:** Sung Choi, Gum-Sung Ryu, Kyeong-Taek Koh, Gi-Hong An, Hyeong-Yeol Kim

**Affiliations:** 1Department of Civil, Architectural, and Environmental Systems Engineering, Sungkyunkwan University (SKKU), 2066, Seobu-Ro, Jangan-Gu, Suwon-Si, Gyeonggi-Do 16419, Korea; csomy1113@naver.com; 2Department of Infrastructure Safety Research, Korea Institute of Civil Engineering and Building Technology, 283, Goyangdae-Ro, Ilsanseo-Gu, Goyang-Si 10223, Korea; ktgo@kict.re.kr (K.-T.K.); agh0530@kict.re.kr (G.-H.A.); hykim1@kict.re.kr (H.-Y.K.)

**Keywords:** alkali-activated material, calcium sulfoaluminate-based expansive additive, concrete shrinkage, modulus of elasticity, shrinkage stress

## Abstract

In this study, a calcium sulfoaluminate-based expansive additive (0%, 2.5%, 5.0%, and 7.5% by the mass of the binder) was added to compensate for the shrinkage of alkali-activated material (AAM) mortar. Modulus of elasticity curves based on the ACI 209 model were derived for the AAM mortar mixed with the additive by measuring the compressive strength and modulus of elasticity. Moreover, autogenous shrinkage and total shrinkage were measured for 150 days, and drying shrinkage was calculated by excluding autogenous shrinkage from total shrinkage. For the autogenous and drying shrinkage of AAM mortar, shrinkage curves by age were obtained by deriving material constants using the exponential function model. Finally, shrinkage stress was calculated using the modulus of elasticity of the AAM mortar and the curves obtained using the shrinkage model. The results showed that the calcium sulfoaluminate-based expansive additive had an excellent compensation effect on the drying shrinkage of AAM mortar, but the effect was observed only at early ages when the modulus of elasticity was low. From a long-term perspective, the shrinkage compensation effect was low when the modulus of elasticity was high, and thus, shrinkage stress could not be reduced.

## 1. Introduction

Cement is an excellent and economical construction material, and to date, no construction material that can perfectly replace cement exists. As environmental problems emerge worldwide, however, efforts are being made to identify a suitable substitute for cement because it emits CO_2_ gas in large quantities [1,2]. It is difficult to completely replace cement in the construction industry because of its widespread usage, but some proportion may be replaceable provided binders suitable for different structural purposes are developed. Various materials that may serve as substitutes for cement have been investigated by many researchers [3,4,5,6,7,8]. Among them, ground granulated blast furnace slag (GGBFS) and fly ash (FA), which have been used to partially replace cement, can exhibit performances equal to those of ordinary Portland cement (OPC) if alkali-activated materials (AAMs) are used. AAMs are eco-friendly materials that can improve the performance of concrete because they have high initial strength and excellent durability. Thus, many researchers have studied the performance of AAM concrete in this regard [9]. Most studies on AAMs, however, have focused on the reactivity and physical performances of binders; the shrinkage characteristics of AAMs have hardly been investigated. Cartwright et al. [10], however, reported that AAMs exhibited 3–6 times more shrinkage than OPC. Various previous studies have also reported that AAMs can cause serious problems when used in structures because their shrinkage is significantly higher than that of OPC [11,12]. However, the mechanisms and inferences relating to the high shrinkage of AAMs have not been comprehensibly summarized.

AAMs cause microcracks because they cause much shrinkage [13,14,15,16], as such microcracks may degrade the strength and durability of concrete [17,18,19,20,21], various methods have been proposed to reduce the shrinkage of AAMs. According to Chatterji [22], the use of the expansive additives containing alkali metal components can compensate for shrinkage because they generate many expansive hydrates, such as ettringite (3CaO·3Al_2_O_3_·CaSO_4_·32H_2_O) and calcium hydroxide (Ca(OH)_2_), and this expansion effect is affected by the mixing and curing conditions of concrete [23,24]. Palacios and Puertas [15] conducted research with various shrinkage-reducing agents to reduce the shrinkage of AAMs. In addition to these methods, which are materials-specific, other techniques for reducing shrinkage have been studied. Thomas et al. [25] proposed a curing method to reduce drying shrinkage, and Sakulich and Bentz [26] reported that use of lightweight aggregates subjected to pre-wetting can reduce autogenous shrinkage due to the internal curing effect. Summarizing the results of these previous studies reveals that the shrinkage caused by AAMs is determined by specific parameters, including the types and mix proportions of AAMs as well as the curing conditions [27,28,29].

With regard to the shrinkage of AAM mortar, structural problems can be caused simply by the generated shrinkage, but most issues are caused by shrinkage cracking, which occurs when the stress caused by shrinkage is higher than the mortar strength. The shrinkage stress acting on a structure increases with the amount of shrinkage and the modulus of elasticity. In general, the shrinkage of concrete is high at early ages and decreases over time, but the modulus of elasticity increases over time. Thus, in the long term, the shrinkage is low, but the shrinkage stress acting on the structure can be evaluated differently. In particular, AAM mortars show higher initial shrinkage than OPC and are also subject to continuous shrinkage over the long term. Therefore, it is necessary to accurately predict the shrinkage stress generated in AAM mortars by measuring the modulus of elasticity by age and to thereafter apply appropriate shrinkage-reducing technologies accordingly.

In this study, a calcium sulfoaluminate-based (CSA) expansive additive was used to compensate for the shrinkage of AAM mortar. The shrinkage characteristics of the AAM mortar were analyzed by age by measuring its autogenous and total shrinkage for 150 days according to the content of the CSA expansive additive, and a shrinkage model was proposed based on the results. In addition, the shrinkage stress of the AAM mortar was calculated by measuring its modulus of elasticity, and the shrinkage stress compensation effect of the mortar mixed with the CSA expansive additive was analyzed.

## 2. Materials and Method

### 2.1. Materials and Mixture Proportions of AAM Mortar

The AAM mortar used in this study contained a two-component binder, wherein GGBFS and FA were mixed in the ratio 7:3. Table 1 shows the analysis results of the major chemical components of the GGBFS and FA. The GGBFS was procured from Sampyo Cement Corp. (Dangjin, Korea). It had a density of 2.91 g/cm^3^ and a fineness of 4683 cm^2^/g. The GGBFS was composed of 41.9% CaO, 33.4% SiO_2_, 13.8% Al_2_O_3_, and 4.9% MgO, and thus, its basicity coefficient (Kb = (CaO + MgO)/(SiO_2_ + Al_2_O_3_)) was 1.00, which is similar to the neutral value of 1.0 for ideal alkali activation [30]. The hydration modulus of GGBFS according to a formula proposed in Ref. [31] (HM = (CaO + MgO + Al_2_O_3_)/SiO_2_) was 1.82. This value was higher than the required value of 1.4, which indicates good hydration properties of the GGBFS [31]. 

The FA was type 2 fly ash (KS L 5405) procured from Sampyo Cement Corp. (Boryeong, Korea) [30]. It had a density of 2.20 g/cm^3^ and a fineness of 3216 cm^2^/g. The FA was composed of 60.3% SiO_2_, 24.2% Al_2_O_3_, and 7.3% Fe_2_O_3_. Moreover, SiO_2_, Al_2_O_3_, and Fe_2_O_3_ accounted for 91.8% of the total, whereas M_2_O (K_2_O + Na_2_O) accounted for 1.9%. The alkali-activator was used to accelerate the reaction of the binder. The alkali-activator was in the form of white powder with a specific density of 1.026 g/cm^3^ and a molar ratio of 0.95. In addition, alkali-activators are manufactured separately by adjusting the chemical components. The SiO_2_/Na_2_O ratio of the alkali-activator (SiO_2_: 46.17%, Na_2_O: 50.18%) used in this study was 0.92. 

To compensate for the shrinkage of the AAM mortar, a powdered CSA expansive additive was used. The CSA expansive additive contained lime, gypsum, and bauxite as its major components. The specific density of the CSA expansive additive was 2.86 g/cm^3^, and its Blaine fineness was 3754 cm^2^/g. River sand with a density of 2.53 g/cm^3^ and a water absorption rate of 1.08% was used as a fine aggregate. The maximum size of the fine aggregate was 4.76 mm, and the fineness modulus was 2.77.

Table 2 summarizes the mix proportions of the AAM mortar. The water-to-binder (W/B) ratio was 45.1% and the sand-to-binder (S/B) ratio was 1.2. The activator-to-water ratio was 24.0%. Then, 0.0%, 2.5%, 5.0%, and 7.5% of CSA expansive additive based on the amount of the binder (GGBFS:FA = 7:3) was added. The AAM mortar was dry mixed for 30 s after inserting the binder as well as the powdered AAM and CSA expansive additive. Water was then added, and the mortar was mixed at a low speed (15 rpm) for 10 min. After inserting the fine aggregate, the mortar was mixed for 90 s at a speed of 30 rpm.

### 2.2. Test Methods

Cube mortar specimens of dimensions 50 mm × 50 mm × 50 mm complying with ASTM C109-16a [32] were prepared. AAM mortar was poured into a cubic mold, cured for 1 d, and demolded. It was then cured in a chamber with constant temperature (20 ± 2 °C) and relative humidity (90 ± 2%). The compressive strength test was conducted at 1, 7, and 28 d of age. The modulus of elasticity of the concrete was calculated by applying loads up to 40% of the ultimate load at a rate of 0.25 MPa/s and obtaining the deformation values for the loads using an interpolation method in accordance with ASTM C469M-14 [33]. 

An embedded gauge (PMFL-50-2LT, Tokyo Sokki Kenkyujo Co., Ltd., Tokyo, Japan) was used to measure the shrinkage of the AAM mortar. Figure 1 shows the method for measuring the length change of the mortar due to shrinkage. An embedded gauge for the length change rate and a temperature sensor (Thermocouple t type, Tokyo Sokki Kenkyujo Co., Ltd., Tokyo, Japan) were installed at the center of 100 mm × 100 mm × 400 mm specimens. Before pouring the mortar, a Teflon sheet and polystyrene board were placed on the inner surface of the mold to minimize friction with the mold and provide restraint in the length direction. After pouring the mortar, a polyester film was installed on the specimen surface to prevent the evaporation and absorption of moisture on the surface. The AAM mortar specimens were cured for 1 d and demolded. The autogenous shrinkage specimen was sealed using a polyester film to control drying shrinkage. The specimen for measuring total shrinkage, namely the sum of drying shrinkage and autogenous shrinkage, was not sealed after demolding. Two specimens were prepared for the length change rate test to measure total shrinkage and autogenous shrinkage, and the measurement results were averaged. The test on the length change rate of the AAM mortar was conducted in a chamber with constant temperature (20 °C) and relative humidity (60%).

## 3. Results and Discussion

### 3.1. Compressive Strength and Modulus of Elasticity

Table 3 shows the tests results of the compressive strength and modulus of elasticity of the AAM mortar according to the content of the expansive additive. The target strength of the AAM mortar was 40 MPa, and all the mixtures met the target strength at 28 d of age. The compressive strength results at 1 d of age showed that the compressive strength of Expansive Additive (EA)-0.0 was 3.16 MPa, but the AAM mortar specimens mixed with the expansive additive exhibited compressive strengths exceeding 5 MPa, indicating a strength increase of more than 2 MPa compared to that for EA-0.0. In particular, EA-5.0 exhibited a compressive strength of 5.91 MPa, the highest observed strength, at 1 d of age. At 28 d of age, the strengths of EA-2.5 and EA-5.0 were respectively 12.6% and 13.7% higher than that of EA-0.0. The strength of EA-7.5, however, was only 8.4% higher. Therefore, it was found that AAM mortar exhibited the highest strength when the content of the expansive additive was 5%.

When the modulus of elasticity of the AAM mortar specimens mixed with the expansive additive were compared, it was found that the initial modulus of elasticity increased as the content of expansive additive increased. While the modulus of elasticity of EA-0.0 at 1 d of age was 1.21 GPa, that of EA-7.5 with the highest expansive additive content was 2.46 GPa, which was approximately two times higher. As the age increased, however, the effect of the addition of the expansive additive on the modulus of elasticity decreased. At 28 d of age, the modulus of elasticity of the AAM mortar ranged from 20.07 to 21.17 GPa, showing that the influence of the expansive additive was not significant.

AAM mortar with the CSA expansive additive exhibited an increase in the modulus of elasticity at early ages, but there was no significant difference in the modulus of elasticity at 28 d of age. This means that the rate of increase of the modulus of elasticity may vary depending on the content of the expansive additive. Various prediction equations on the modulus of elasticity were applied to compare the moduli of elasticity of the AAM mortar according to the content of the expansive additive, but the model on the modulus of elasticity proposed by American Concrete Institute (ACI) 209, which can predict the modulus of elasticity of the AAM mortar most accurately, was used [34]. As a result, R^2^ between the modulus of elasticity measured from the experiment and that obtained by the ACI 209 model for AAM mortar was 98.89% or higher, indicating a high correlation.
(1)$$E_{cmt}=E_{cm28}\sqrt{\frac{t}{a+bt}}$$
where *E_cmt_* is the modulus of elasticity of the AAM mortar at *t* days of age, and *E*_*cm*28_ is the compressive strength at 28 d of age. *a* and *b* are material constants related to the compressive strength. Figure 2 shows the experimental values of the modulus of elasticity of the AAM mortars and the prediction curves for the modulus of elasticity as per the ACI 209 model. Table 4 shows the results of the derivation of the values of *a* and *b* according to the content of the expansive additive by applying the ACI 209 model. Nagataki and Gomi [35] reported that if the content of CSA expansive additive exceeds a certain value, the strength may decrease but the modulus of elasticity and creep increase continuously. The experiment results also showed that the material constants of the modulus of elasticity did not change considerably when the content of the expansive additive was less than 5%, but they exhibited a different tendency when the content was 7.5% because the increment in the modulus of elasticity increased. When the content of the expansive additive was less than 5%, the value of *a*, which represents the increment in the modulus of elasticity, decreased, whereas the value of *b*, which denotes the rate of increase in the modulus of elasticity, showed a tendency to slowly increase as the content of the expansive additive increased at 1 d of age. This means that when the expansive additive is added, the initial modulus of elasticity is high and the modulus of elasticity rapidly increases, but there is no significant difference in the final modulus of elasticity. As a result, the prediction curves for the modulus of elasticity as per the ACI 209 model showed a tendency similar to the actually measured modulus of elasticity. Regarding the curves for the modulus of elasticity predicted by the ACI 209 model, EA-7.5 exhibited a somewhat high modulus of elasticity in the long term, but the AAM mortar specimens with the expansive additive content of less than 5% exhibited no significant difference in the modulus of elasticity.

### 3.2. Shrinkage

Figure 3 shows the shrinkage test results of the AAM mortars at 3 and 150 d of age according to the content of the expansive additive. In the AAM mortar shrinkage test, measurement was started based on the final setting time. According to the results of various studies on AAMs, OPC-based mortar has large initial autogenous shrinkage, which tends to decrease over time. However, AAM mortar has higher initial autogenous shrinkage than OPC mortar and involves high shrinkage in the long term [11,36]. In this experiment, the shrinkage test results of the AAM mortars also showed that rapid shrinkage occurred until approximately 0.5 d and continuous shrinkage occurred until 150 d. EA-0.0 exhibited the highest total shrinkage and autogenous shrinkage because it had no shrinkage compensation effect caused by the expansive additive. As the content of the expansive additive increased, the shrinkage of the AAM mortar decreased. The shrinkage compensation effect of the expansive additive was most clearly observed within 1 d of age. Moreover, the shrinkage curves for up to 150 d of age showed that the shrinkage compensation effect of the expansive additive was not significant for autogenous shrinkage, but the opposite was true for total shrinkage.

The total shrinkage of AAM mortar is the sum of autogenous and drying shrinkage. Thus, drying shrinkage can be calculated using the measured total and autogenous shrinkage. Table 5 summarizes the autogenous, total, and drying shrinkage of the AAM mortars at 1 and 150 d of age. When the shrinkage of AAM mortars was analyzed at 1 d of age, the autogenous shrinkage reduction rates of EA-2.5, EA-5.0, and EA-7.5 were 23.3%, 27.0%, and 35.3% respectively compared to the autogenous shrinkage of EA-0.0, and their corresponding drying shrinkage reduction rates were 65.0%, 65.9%, and 85.1%. This indicates that the addition of expansive additive to AAM mortar reduces both autogenous and drying shrinkage at 1 d of age, but the influence on drying shrinkage is higher. When the shrinkage of the AAM mortars was analyzed at 150 d of age, the autogenous shrinkage reduction rates of EA-2.5, EA-5.0, and EA-7.5 were 1.8%, 3.9%, and 7.5% respectively compared to the autogenous shrinkage of EA-0.0; thus, the reduction rates were lower compared to those at 1 d of age. The drying shrinkage reduction rates of EA-2.5, EA-5.0, and EA-7.5 were 38.0%, 67.4%, and 71.8% respectively compared to the drying shrinkage of EA-0.0, indicating that the reduction rates were lower compared to those at 1 d of age even though shrinkage was reduced by the expansive additive in the long term. This result can be attributed to the initial shrinkage compensation effect of the CSA expansive additive, and the shrinkage reduction rate decreased in the long term because the shrinkage compensation effect of the expansive additive was not significant after 3 d of age.

### 3.3. Shrinkage Modeling

The shrinkage of concrete and mortar is affected by various mixing and curing conditions, and it must be measured for a long period of time. Therefore, various models have been proposed to predict shrinkage in advance. The reaction mechanisms of the AAMs, however, are different from those of the existing OPC-based mortar and concrete. They are also characterized by different binder reaction times and reaction rates. Therefore, a model that reflects the effects of materials must be selected to predict the shrinkage of the AAM mortar using a model. Hu et al. [37] applied various models to predict the autogenous and drying shrinkage of alkali-activated slag mortar and examined their suitability. They also reported that the exponential function model is suitable for predicting the autogenous and drying shrinkage of AAM mortar. In the exponential function model, the effects of materials and mixing can be reflected through the values of the constants. Thus, an exponential function model was applied to analyze the shrinkage of the AAM mortars mixed with the expansive additive, as follows:(2)$$\varepsilon_{\left(t\right)}=\varepsilon_{s}\left[1-\alpha \cdot \exp\left(-bt\right)\right]$$

In this exponential model, *ε*_(*t*)_ is the shrinkage at *t* days of age and *ε*_150_ is the shrinkage at 150 d of age. Both α and *b* are material constants, where α denotes the generated amount of shrinkage and *b* is the slope of generated shrinkage over time, and subscripts _a_ and _d_ refer to the autogenous shrinkage and drying shrinkage respectively. Table 6 summarizes the material constants and amount of shrinkage at 150 d of age for the AAM mortars mixed with the expansive additive using the exponential model. When the material constants for the autogenous shrinkage of the AAM mortars were analyzed, the value of α_a_ for EA-0.0 was −0.499, while those for EA-2.5, EA-5.0, and EA-7.5, which used an expansive additive, ranged from −0.553 to −0.572. Thus, the values of α_a_ for the AAM mortar specimens that used the expansive additive were lower. The values of *b*_a_, however, showed no significant difference despite a slight increase when the expansive additive was added. Moreover, when the material constants for the drying shrinkage of the AAM mortars were analyzed, it was found that the value of α_d_ for EA-0.0 was −0.485, but those for the AAM mortar specimens that used the expansive additive ranged from −0.700 to −0.944. Thus, the use of the expansive additive significantly decreased the value of α_d_. The value of *b*_d_, however, exhibited no significant difference, similar to the case of autogenous shrinkage. 

Figure 4 shows the relationships between the expansive additive content and the coefficients of the shrinkage model. While the material constants of autogenous shrinkage did not exhibit significant changes even when the expansive additive content increased, α_d_ significantly decreased as the expansive additive content increased. This indicates that the CSA expansive additive has a larger impact on drying shrinkage than on autogenous shrinkage. An increase in the expansive additive content can compensate for and reduce shrinkage, but this mostly results from the expansion effect at early ages. The value of *b* was not affected by the expansive additive content. This is because the shrinkage reduction effect of the expansive additive was not significant in a long term.

### 3.4. Shrinkage Stress

To calculate the shrinkage stress of the AAM mortars, the prediction curves for the modulus of elasticity obtained by the ACI 209 model and the total shrinkage prediction curves of the AAM mortars obtained by the exponential function were used. The shrinkage stress of the AAM mortars (f_(*sh*, *t*+__Δ*t*)_) was calculated by adding the stress caused by the shrinkage generated per unit time (Δ*f**_sh_*) to the shrinkage stress acting on the AAM mortar (*f*_(*sh*, *t*)_). 

*f*_(*sh*, *t*+__Δ*t*)_ = *f*_(*sh*, *t*)_ + Δ*f**_sh_*,(3)

Δ*f*_*sh*_ = *E*·Δ*ε*_*sh*_.(4)

Figure 5 shows the stress generated by shrinkage at 30 min intervals (Δ*f**_sh_*) and the shrinkage stress accumulated in the AAM mortars (*f_sh_*) over time. The shrinkage stress generated per unit time in Figure 5a shows that a large amount shrinkage occurred until 1 d of age, and thus, Δ*f**_sh_* was also high even though the modulus of elasticity was low in that period. Δ*f**_sh_* rapidly decreased after 1 d of age, but it showed a tendency to slowly increase as the age increased. This is because the modulus of elasticity increased with the age. For the AAM mortars, shrinkage stress continuously occurred until 150 d as total shrinkage increased. Δ*f**_sh_* mostly ranged from 0.001–0.003 MPa, but it showed a higher range (0.004–0.006 MPa) within 60 d of age. Large shrinkage stress was observed in some sections because some differences were observed in Δ*ε**_sh_* when the shrinkage obtained by the data logger was divided into 30 min intervals. The increase in the expansive additive content decreased the maximum value of Δ*f**_sh_* at 1 d of age, but the expansive additive content could not significantly affect Δ*f**_sh_* after 1 d of age. 

As shown in Figure 5b, the shrinkage stress accumulated in the AAM mortars (*f_sh_*) was hardly affected by the expansive additive content. At early ages, the shrinkage stress decreased as the expansive additive content increased. After 6 d of age, however, such tendency was not observed. For the AAM mortars, the shrinkage stress continuously increased until 150 d of age. Table 7 summarizes the cumulative shrinkage stress at 1, 28, and 150 d of age. The shrinkage stress at 1 d of age ranged from 0.54 to 0.26 MPa, which was approximately 12.1–6.1% of the shrinkage stress at 150 d of age. The shrinkage stress at 28 d of age ranged from 2.37 to 1.90 MPa, which was approximately 55.5–43.5% of the shrinkage stress at 150 d of age. These results show that relatively larger shrinkage stress occurred at early ages. After 28 d of age, however, continuous shrinkage stress was observed in the AAM mortars. This value increased until 150 d without reduction even when the expansive additive was added. When the shrinkage stress values according to the expansive additive content at 1 and 150 d of age were compared, it was found that the shrinkage stress of the AAM mortars reduced by 29.6–51.9% at 1 d of age depending on the expansive additive content. However, the shrinkage stress reduction rate decreased to less than 5% at 150 d of age. This is because the expansive additive could not reduce the shrinkage stress in the long term as it reacted at early ages, resulting in an expansion, and could not generate further expansion thereafter. 

## 4. Conclusions

The compressive strength, moduli of elasticity, and shrinkage test results of AAM mortars were obtained according to the content of the CSA expansive additive, and a shrinkage model was derived based on the results. In addition, the following conclusions were drawn from calculations of the shrinkage stress.

The addition of CSA expansive additive increased the compressive strength and modulus of elasticity of the AAM mortars at 1 d of age, and this tendency increased as the CSA expansive additive content increased. After 1 d of age, however, the addition of CSA expansive additive could hardly improve the compressive strength and moduli of elasticity of the AAM mortars.The AAM mortars exhibited autogenous shrinkage to a greater extent than drying shrinkage, but the CSA expansive additive had a larger impact on reducing drying shrinkage. Therefore, limitations exist with regard to controlling the shrinkage of AAM mortar with the CSA expansive additive alone, and it is necessary to apply additional methods for controlling autogenous shrinkage.A model capable of predicting the modulus of elasticity and shrinkage of AAM mortar by age was proposed, and the shrinkage stress of the AAM mortar mixed with the CSA expansive additive at 150 d of age was calculated using the proposed model. The result showed that the CSA expansive additive was suitable for controlling the cracks caused by the shrinkage of the AAM mortar at early ages, but it caused no improvement with regard to shrinkage generated in the long term. Therefore, to ensure effective control of the shrinkage stress of AAM mortar, it is necessary to accurately quantify the stress through long-term shrinkage stress monitoring and to apply techniques capable of controlling shrinkage stress properly depending on the time of shrinkage occurrence.The results of this study in predicting cracks caused by the shrinkage of AAM mortar will be used for material improvement and curing management methods to minimize the shrinkage of AAM.

## Figures and Tables

**Figure 1 materials-12-03312-f001:**
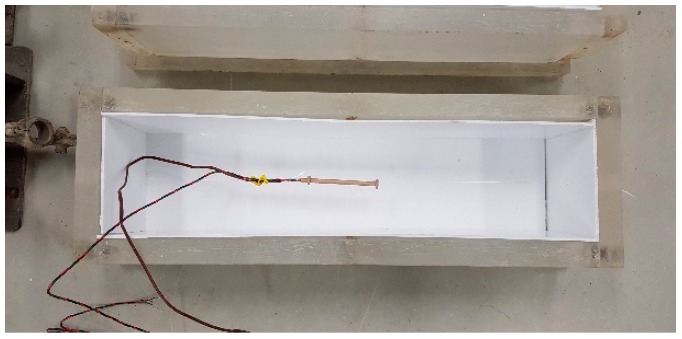
Measurement of length change.

**Figure 2 materials-12-03312-f002:**
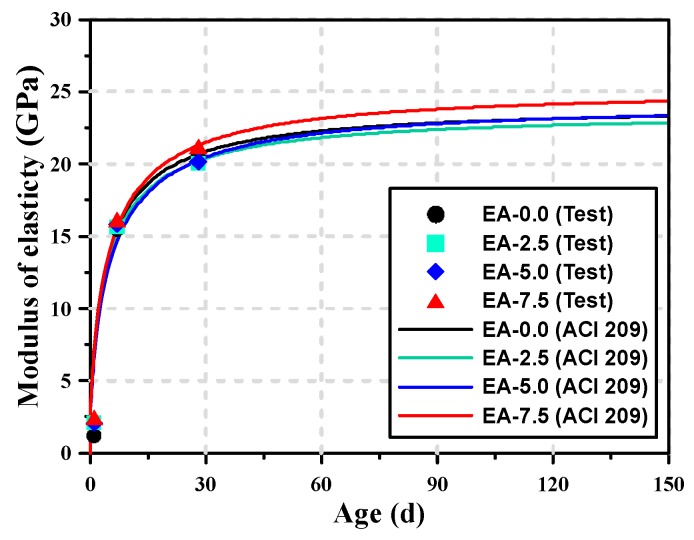
Comparison of modulus of elasticity between the test results and the results of the ACI 209 model.

**Figure 3 materials-12-03312-f003:**
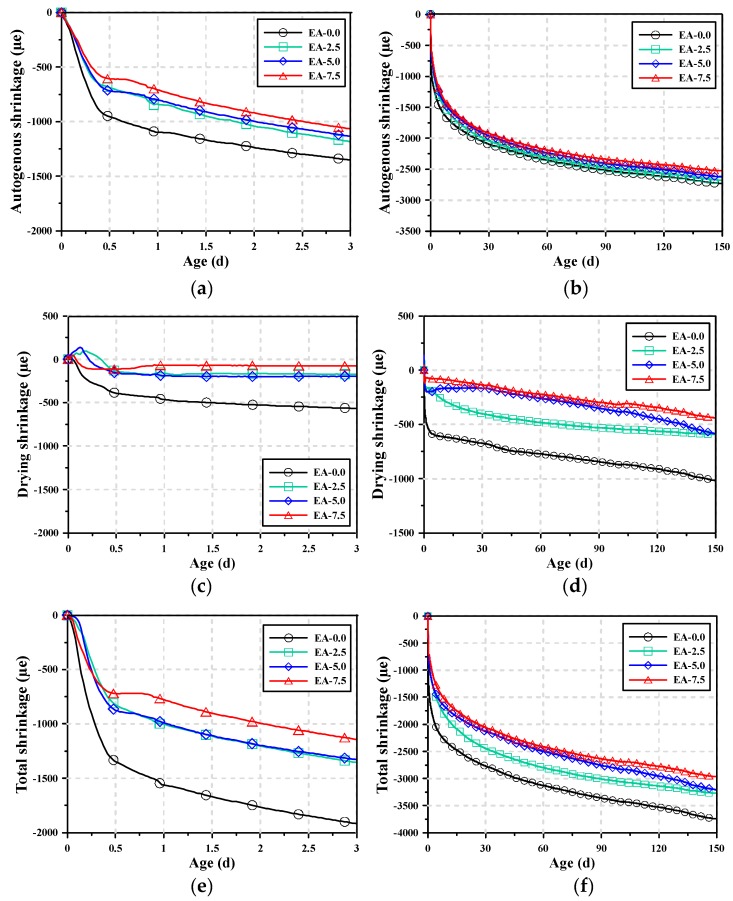
Autogenous shrinkage curves of AAM mortars: (**a**) autogenous shrinkage for 3 d, (**b**) autogenous shrinkage for 150 d, (**c**) drying shrinkage for 3 d, (**d**) drying shrinkage for 150 d, (**e**) total shrinkage for 3 d, and (**f**) total shrinkage for 150 d.

**Figure 4 materials-12-03312-f004:**
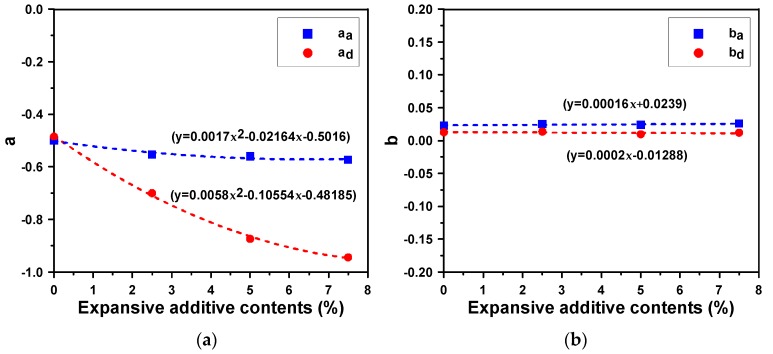
Relationships between the expansive additive content and material coefficients of the shrinkage model: (**a**) for coefficient “α”, and (**b**) for coefficient “*b*”.

**Figure 5 materials-12-03312-f005:**
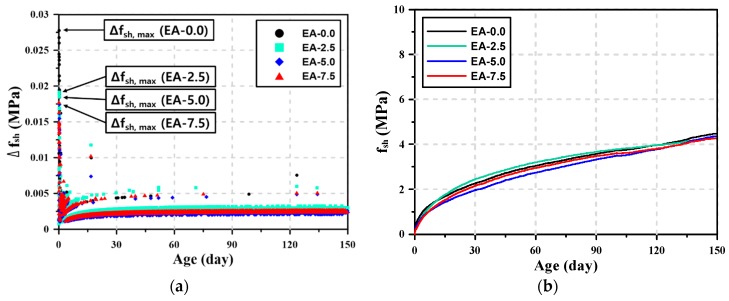
Stress generated by shrinkage. (**a**) Stress generated per unit time (Δ*f_sh_*), and (**b**) shrinkage stress (*f_sh_*).

**Table 1 materials-12-03312-t001:** Chemical composition of ground granulated blast furnace slag (GGBFS), fly ash (FA), and calcium sulfoaluminate-based (CSA) expansion agent.

Type	CaO	SiO_2_	Al_2_O_3_	Fe_2_O_3_	SO_3_	MgO	K_2_O	Na_2_O
GGBFS	41.9	33.4	13.8	0.6	4.8	4.9	0.5	0.2
FA	3.7	60.3	24.2	7.3	0.5	1.9	1.2	0.8
CSA	36.4	30.2	24.2	1.7	5.3	1.4	0.5	0.3

**Table 2 materials-12-03312-t002:** Mix proportions of alkali-activated material (AAM) mortar.

Type	W/B	S/B	Water (g)	Binder (g)	Activator (g)	Sand (Fine Aggregate) (g)	CSA Expansive Additive (g)
EA-0.0	0.451	1.2	451	1000	108	1200	0
EA-2.5	0.451	1.2	451	1000	108	1200	25
EA-5.0	0.451	1.2	451	1000	108	1200	50
EA-7.5	0.451	1.2	451	1000	108	1200	75

**Table 3 materials-12-03312-t003:** Compressive strength and modulus of elasticity of AAM mortar.

Type	Compressive Strength (MPa)			Modulus of Elasticity (GPa)		
	1 d	7 d	28 d	1 d	7 d	28 d
EA-0.0	3.16	33.96	43.59	1.21	15.49	20.53
EA-2.5	5.18	37.70	49.09	2.13	15.67	20.07
EA-5.0	5.91	35.22	49.56	2.07	15.84	20.16
EA-7.5	5.68	34.98	47.26	2.46	16.15	21.17

**Table 4 materials-12-03312-t004:** Material constants of the modulus of elasticity of AAM mortar.

Type	EA-0.0	EA-2.5	EA-5.0	EA-7.5
*a*	8.4	7.4	7.3	8.0
*b*	0.689	0.722	0.727	0.703
*R* ^2^	0.9915	0.9895	0.9889	0.9914

**Table 5 materials-12-03312-t005:** ε_ash*_, ε_tsh**_, ε_dsh***_ of AAM mortar at 1 and 150 d of age.

Type	ε_ash_ (με)		ε_tsh_ (με)		ε_dsh_ = ε_tsh_ − ε_ash_ (με)	
	1 d	150 d	1 d	150 d	1 d	150 d
EA-0.0	−1096	−2330	−1559	−3096	−463	−766
EA-2.5	−859	−2290	−1003	−2765	−162	−475
EA-5.0	−800	−2211	−958	−2461	−158	−250
EA-7.5	−709	−2165	−778	−2381	−69	−216

* ε_ash_: Autogenous shrinkage, ** ε_tsh_: Total shrinkage, *** ε_dsh_: Drying shrinkage.

**Table 6 materials-12-03312-t006:** Material constants of the exponential function model for autogenous and drying shrinkage.

Type	Autogenous Shrinkage				Drying Shrinkage			
	α_a_	*b* _a_	ε_as, 150_	*R* ^2^	α_d_	*b* _d_	ε_ds, 150_	*R* ^2^
EA-0.0	−0.499	0.023	3745.2	0.9574	−0.485	0.0127	−1016.9	0.9977
EA-2.5	−0.553	0.025	3270.1	0.9566	−0.700	0.0136	−591.3	0.9752
EA-5.0	−0.560	0.024	3207.8	0.9605	−0.874	0.0097	−587.2	0.9575
EA-7.5	−0.572	0.026	2963.5	0.9565	−0.944	0.0121	−441.0	0.9521

**Table 7 materials-12-03312-t007:** Shrinkage stress by age according to expansive additive content.

Type	*f_sh_* (MPa)			Increase/Decrease Rate Compared to the Reference (EA-0.0) (%)		
	1 d	28 d	150 d	1 d	28 d	150 d
EA-0.0	0.54	2.20	4.48	0.0	0.0	0.0
EA-2.5	0.38	2.37	4.27	−29.6	+7.7	−4.7
EA-5.0	0.37	1.90	4.37	−31.5	−13.6	−2.5
EA-7.5	0.26	2.08	4.27	−51.9	−5.5	−4.7

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
