# Peer review of "Experimental Study on the Shrinkage Behavior and Mechanical Properties of AAM Mortar Mixed with CSA Expansive Additive"

_materials, 2019, doi:10.3390/ma12203312_

Round 1

Reviewer 1 Report

The manuscript reports the results of an experimental study describing the impact of the addition of an expansive admixture to AAM.

While clearly presented the manuscript reads more like a lab report than a scientific paper - that is it focuses on the reporting of the observations rather than providing a deeper understanding of the mechanisms and implications of what is observed. For this reason changes are recommended such that the authors can better describe the fundamental contributions of the work, without this deeper analysis I would recomend against publication.

The manuscript requires some English language editing.

Author Response

September 26, 2019

Dear reviewer,

Thank you very much for your valuable comments. I, with my co-authors, carefully revised the manuscript entitled “Experimental Study on the Shrinkage Behavior and Mechanical Properties of AAM mortar mixed with CSA Expansive Additive” according to the reviewer’s comments as follows.

Comment 1: While clearly presented the manuscript reads more like a lab report than a scientific paper - that is it focuses on the reporting of the observations rather than providing a deeper understanding of the mechanisms and implications of what is observed. For this reason changes are recommended such that the authors can better describe the fundamental contributions of the work, without this deeper analysis I would recomend against publication.

Response 1: Mechanisms for the binding reactions of AAM have not been clearly identified, and many researchers have attempted to identify the reaction mechanisms of AAM using various methods and techniques. Therefore, this study aimed to present and verify an evaluation method for the shrinkage stress and cracking using the relationship between the load-carrying capacity and the external stress caused by shrinkage to address the cracking problem caused by shrinkage that occurs during the reaction and hardening processes of AAM.

This study presented a model for predicting the shrinkage and modulus of elasticity of AAM mortar according to the used amount of the CSA expansive additive and the age as well as a method for evaluating the shrinkage stress. This methods is different from the existing approaches. When the CSA expansive additive, which is evaluated to have an excellent shrinkage compensation effect for OPC, was applied to AAM mortar through this shrinkage stress evaluation method, it was found that the CSA expansive additive was not effective in reducing the shrinkage stress of AAM mortar. Moreover, it was verified in this study that the CSA expansive additive alone has limitations in controlling the shrinkage of AAM mortar because the CSA expansive additive is more effective in reducing drying shrinkage while AAM mortar is more subjected to autogenous shrinkage than drying shrinkage. Based on these results, we are planning to develop research on materials and curing methods that simply reduce only shrinkage into research on verifying the effect of applying this shrinkage monitoring technique on the crack reduction in mortar and concrete.

Reflecting the opinions of the reviewer, however, our intention has been clearly described in the section of conclusions. We will clearly state our intentions in the concluding section by reflecting the reviewer's opinions.

“4. The results of this study in predicting cracks caused by the shrinkage of AAM mortar will be used for material improvement and curing management methods to minimize the shrinkage of AAM.” [Line 350-352]

Comment 2: The manuscript requires some English language editing.

Response 2: Some awkward sentences have been modified.

1) “Form” has been modified to “From”[Line 30]

2) “The activator-to-water ratio was 24.0%” [Line 113]

Reviewer 2 Report

The authors present a research focused on the shrinkage behavior of alkali activated mortars with calcium sulfoaluminate-based additive. The results showed that the calcium sulfoaluminate-based expansive additive had an excellent compensation effect on the drying shrinkage of AAM mortar, but the effect was observed only at early ages when the modulus of elasticity was low.

The topic is up to date, the used methods are appropriate, the organisation of the manuscript is clear, the lot of the experimental results are presented. The conclusions are based on the obtained findings. I have only several comments and recommendations to be considered:

Abstract - please check: "Form a long-term perspective..." or "From a long-term perspective..."? Table 1 - The unit is missing in the Table, I guess %, add that please. In my opinion, the water-to-binder ratio could be better expressed as 0.451 instead of percentage expression.

Author Response

September 26, 2019

Dear reviewer,

Thank you very much for your valuable comments. I, with my co-authors, carefully revised the manuscript entitled “Experimental Study on the Shrinkage Behavior and Mechanical Properties of AAM mortar mixed with CSA Expansive Additive” according to the reviewer’s comments as follows.

Comment 1: Abstract - please check: "Form a long-term perspective..." or "From a long-term perspective..."?

Response 1: “Form” has been modified to “From”.[Line 30]

Comment 2: Table 1 - The unit is missing in the Table, I guess %, add that please. In my opinion, the water-to-binder ratio could be better expressed as 0.451 instead of percentage expression.

Response 2: In Table 1, W/B was modified to 0.451. and S/B is the ratio of sands to binders, and there is no unit. [Line 119]

Reviewer 3 Report

The paper deals with the use of CSA-based expansive agent to mitigate the shrinkage of alkali-activated slag-based mortars. In particular, several tests were carried out by the authors by varying the CSA dosage.

The paper is well written and well structured, the topic is interesting.

Anyway, I have some comments about the paper:

Several information about the materials used in this experimentation are missing. For example, no information about activators are provided to the reader, fineness of expansive agent is missing and maximum size of sand is not reported in the text. Authors did not specify the number of specimens subjected to test. This is fundamental for evaluating the soundness of the results. Information about fresh properties of mortars are missing. In particular, it is well known that the addition of expansive agents could lead to a strong reduction in workability and variations in setting times. These aspects cannot be neglected. Are the authors sure to use the ACI209 model to predict the elastic modulus of AAS? The approach used to evaluate the shrinkage stress is very interesting. Anyway, in order to predict the crack formation, it could be necessary to study also the tensile strength development over time. When shrinkage stress is equal to the tensile strength, the mortars can crack. The references are often more than 15 years old. Please, see the suggestion reported in the attached PDF.

Finally, other minor comments can be found in the attached PDF.

For the abovementioned reasons, the paper can be accepted after minor revision.

Author Response

September  26, 2019

Dear reviewer,

Thank you very much for your valuable comments. I, with my co-authors, carefully revised the manuscript entitled “Experimental Study on the Shrinkage Behavior and Mechanical Properties of AAM mortar mixed with CSA Expansive Additive” according to the reviewer’s comments as follows.

 Comment 1: Information about fresh properties of mortars are missing. In particular, it is well known that the addition of expansive agents could lead to a strong reduction in workability and variations in setting times.

Response 1: Although there are measurement results for the flow of AAM mortar as follows, contents on workability have not been included in the paper because this study was conducted with focus on the modulus of elasticity and length change rate of hardened mortar. The final setting time of AAM mortar is important for determining the time of measuring the length change rate, and we measured the length change rate based on the final setting time obtained through the setting time test.

“In the AAM mortar shrinkage test, measurement was started based on the final setting time.”[Line 203]

EA-0.0

EA-2.5

EA-5.0

EA-7.5

Flow (mm)

212

187

173

169

Initial setting time (min)

101

93

70

55

Final setting time (min)

292

238

223

171

Comment 2: Are the authors sure to use the ACI 209 model to predict the elastic modulus of AAS?

Response 2: Among the models for predicting the modulus of elasticity over time using the modulus of elasticity at the age of 28 days, the most influential models are ACI 209 and Eurocode2. We chose the ACI 209 model, and the value measured from the experiment and value obtained by the ACI 209 model exhibited R2 of 98.8% or higher, indicating a high correlation, as expressed in the manuscript.

“As a result, R2 between the modulus of elasticity measured from the experiment and that obtained by the ACI 209 model for AAM mortar was 98.89% or higher, indicating a high correlation.” [Line 171-173]

Comment 3: In order to predict the crack formation, it could be necessary to study also the tensile strength development over time. When shrinkage stress is equal to the tensile strength, the mortars can crack.

Response 3: Research on the evaluation of cracks caused by the shrinkage stress will be conducted.

Comment 4: The references are often more than 15 years old

Response 4: The latest references below have been added.

Coppola, L.; Coffetti, D.; Crotti, E. & Pastore, T. CSA-based Portland-free binders to manufacture sustainable concretes for jointless slabs on ground. 2018, Construction and Building Materials, 187. pp 691-698. [Line 362-363]

Coppola, L.; Coffetti, D.; Crotti, E.; Marini, A.; Passoni, C. & Pastore, T. Lightweight cement-free alkali-activated slag plaster for the structural retrofit and energy upgrading of poor quality masonry walls. 2019, Cement and Concrete Composites, 104, 103341. [Line 375-377]

Coppola, L.; Coffetti, D. & Crotti, E. Pre-packed alkali activated cement-free mortars for repair of existing masonry buildings and concrete structures. 2018, Construction and Building Materials, 173, pp 111-117. [Line 384-385]

Comment 5: What does this sentence mean? “AAM was used to accelerate the reaction of the binder.”

Response 5: “AAM” has been modified to “the alkali-activator”.

“The alkali-activator was used to accelerate the reaction of the binder.” [Line 100-101]

Comment 6: Please, provide information about the activators (sodium silicate, sodium hydroxide or other chemical compound)

Response 6: Information on the alkali-activator has been added.

“The alkali-activator was in the form of white powder with a specific density of 1.026 g/㎤ and a molar ratio of 0.95. In addition, alkali-activators are manufactured separately by adjusting the chemical components. The SiO2/Na2O ratio of the alkali-activator (SiO2 : 46.17%, Na2O : 50.18%) used in this study was 0.92.” [Line 101-104]

Comment 7: Please, provide the specific surface in order to evaluate the reactivity of the powder. “CSA expansive”

Response 7: Information on the CSA expansive additive has been added.

“The specific density of the CSA expansive additive was 2.86 g/㎤, and its Blaine fineness was 3,754 ㎠/g” [Line 107-108]

Comment 8: Please, specify the maximum size of sand.

Response 8: The maximum size of the fine aggregate was 4.76 mm. The corresponding text has been modified.

“The max. size of the fine aggregate was 4.76 mm” [Line 109]

Comment 9: How did you evaluate the chemical composition of binders and additives? XRF or other? “Table 1. Chemical composition of GGBFS, FA, and CSA expansion agent“

Response 9: The chemical compositions of the binder and CSA expansive additive were evaluated using XRF test equipment (BRUKER S8 Tiger, Billerica, MA, USA). [Line 110]

Comment 10: The sentence is not clear.” The proportion of AAM in a unit quantity was 24%.”

Response 10: The sentence has been modified.

“The activator-to-water ratio was 24.0%” [Line 113]

Comment 11: the space is missing.”binder(GGBFS:FA=7:3)”

Response 11: It has been modified.

“binder (GGBFS:FA=7:3)” [Line 114-115]

Comment 12: Pastes are generally composed by binders, water and admixtures. If you add to the mixture the sand, the correct name is mortar.”AAS pastes”

Response 12: It has been modified to “AAM mortar”. [Line 119]

Comment 13: How many specimens did you test for mechanical and physical characterization of mortars?. ”The test on the length change rate of the AAM mortar was conducted in a chamber with constant temperature (20 °C) and relative humidity (60%).”

Response 13: Two specimens were prepared for the length change rate test, and the measurement results of each specimen were averaged.

“Two specimens were prepared for the length change rate test to measure total shrinkage and autogenous shrinkage, and the measurement results were averaged.” [Line 138-140]

Comment 14: How can you use the ACI209 model to predict the elastic modulus of AAS by using only three experimental data? Moreover, are you sure that the ACI209 model is usefoul for alkali activated materials? ”ACI 209 was applied”

Response 14: We applied the ACI 209 model to express the modulus of elasticity of AAM mortar as a function of time. The ACI 209 model has coefficients (a and b) that consider the influence of the material in predicting the modulus of elasticity, and we applied the values of the coefficients that produced a result close to the modulus of elasticity of the AAM mortar mixed with the expansive additive measured from the experiment. The error rate (R2) between the experimental value and that of the model was 98.89% or higher.

Comment 15: This is an interesting approach to the shrinkage. Anyway, in order to predict the crack formation, it could be interesting to study also the tensile strength development over time. When shrinkage stress is equal to the tensile strength, the mortars can crack. “3.4 Shrinkage stress”

Response 15: Further research will be conducted to present a method for reducing the cracks of AAM mortar by identifying the crack occurrence time due to shrinkage through the relationship between the tensile strength and shrinkage stress of AAM mortar and by applying various methods to reduce the shrinkage of mortar.

 [오전1]Thank you for using this service. As promised to you, this file has only undergone the translation stage of the service and has not yet undergone language editing. Please go through this file to ensure that the translation conveys your intended meaning. Note that any actions for revising the language, grammar, or structure of this text will be done in the editing stage. We hope to receive this file from you in order to start the editing.

Round 2

Reviewer 1 Report

Manuscript can be accepted in current form.